# A Comprehensive Overview of Tomato Processing By-Product Valorization by Conventional Methods versus Emerging Technologies

**DOI:** 10.3390/foods12010166

**Published:** 2022-12-29

**Authors:** Elham Eslami, Serena Carpentieri, Gianpiero Pataro, Giovanna Ferrari

**Affiliations:** 1Department of Industrial Engineering, University of Salerno, Via Giovanni Paolo II, 132, 84084 Fisciano, Italy; 2ProdAl Scarl, Via Giovanni Paolo II, 132, 84084 Fisciano, Italy

**Keywords:** tomato by-products, biorefinery, conventional solvent extraction, novel technologies, bioactive compounds, biofuels

## Abstract

The tomato processing industry can be considered one of the most widespread food manufacturing industries all over the world, annually generating considerable quantities of residue and determining disposal issues associated not only with the wasting of invaluable resources but also with the rise of significant environmental burdens. In this regard, previous studies have widely ascertained that tomato by-products are still rich in valuable compounds, which, once recovered, could be utilized in different industrial sectors. Currently, conventional solvent extraction is the most widely used method for the recovery of these compounds from tomato pomace. Nevertheless, several well-known drawbacks derive from this process, including the use of large quantities of solvents and the difficulties of utilizing the residual biomass. To overcome these limitations, the recent advances in extraction techniques, including the modification of the process configuration and the use of complementary novel methods to modify or destroy vegetable cells, have greatly and effectively influenced the recovery of different compounds from plant matrices. This review contributes a comprehensive overview on the valorization of tomato processing by-products with a specific focus on the use of “green technologies”, including high-pressure homogenization (HPH), pulsed electric fields (PEF), supercritical fluid (SFE-CO_2_), ultrasounds (UAE), and microwaves (MAE), suitable to enhancing the extractability of target compounds while reducing the solvent requirement and shortening the extraction time. The effects of conventional processes and the application of green technologies are critically analyzed, and their effectiveness on the recovery of lycopene, polyphenols, cutin, pectin, oil, and proteins from tomato residues is discussed, focusing on their strengths, drawbacks, and critical factors that contribute to maximizing the extraction yields of the target compounds. Moreover, to follow the “near zero discharge concept”, the utilization of a cascade approach to recover different valuable compounds and the exploitation of the residual biomass for biogas generation are also pointed out.

## 1. Introduction

The food supply chain, including production, processing, consumption, and disposal, generates waste along the different steps, from harvesting and industrial processing to storage, distribution, and households. The food processing industry is responsible for producing a considerable quantity of residues and by-products, making it the second-largest producer after households [1].

In particular, among all food categories, fruits and vegetables are actually the most wasted (45% of the total amount of food globally wasted) [2]. Interestingly, data shows that almost half of all fruits and vegetables produced are discarded along the supply chain [3]. Fruit and vegetable processing by-products, mostly consisting of pomace (peel, pulp, and seeds), are excellent sources of ingredients such as dietary fibers, peptides, proteins, and polysaccharides, as well as bioactive compounds such as polyphenols, antimicrobial compounds, and natural pigments [4].

Currently, the food and beverage industry has been facing a major challenge in handling and disposing of processing wastes and by-products while striving to address the growing consumers’ demand for natural health-beneficial ingredients. To face these challenges, food waste valorization and their transformation into valuable products are widely investigated worldwide [5].

Among fruits and vegetables, tomato is one of the most widespread crops in the world, with a global annual production that exceeded 180 million tons in 2021 [6]. Italy, with a production that, in 2022, reached about 5.5 million tons, accounts for 53% of the total European production, with the Campania Region being the largest Italian production area [7]. The top ten countries for tomato production are illustrated in Figure 1 [8].

During tomato processing, huge quantities of by-products are generated, accounting for 3–5% (*w*/*w*) of the total raw tomatoes [9,10]. They consist of peels, seeds, and a small amount of pulp, named tomato pomace, which causes the main disposal issue for the tomato processing industry. Currently, tomato pomace has low-added-value and is mainly used as animal feed, to produce compost, or is discharged in landfills, creating environmental loads [11]. However, many studies are showing that tomato processing by-products are a rich source of fiber [12,13], polyphenols [14,15], carotenoids (such as lycopene and β-carotene) [11,16,17,18], oil [19], and proteins [15], which can be very beneficial to human nutrition and health due to their chemical characteristics and biological potential. Thus, the recovery of high-added-value compounds from tomato processing by-products could represent a strategic option that is extremely beneficial for the tomato processing industry and consumers and very valuable from an environmental perspective [17]. However, in order to recover bioactives, which are mainly intracellular compounds, effective extraction processes are required [20].

Generally, high-added-value compounds are firmly locked in the plant cells, with the cell envelopes representing a physical barrier to mass transfer, which could hinder their recovery via conventional solvent extraction (CSE). For decades, conventional extraction methods such as maceration and reflux extraction have been utilized. However, these methods suffer from several drawbacks, such as low yield, long extraction times, requirement of high amounts of organic solvents, isomerization, and degradation of some compounds when higher processing temperatures are utilized, all of which affect processing costs while also exacerbating environmental concerns [21]. To overcome these drawbacks, the application of novel technologies weakening the cell membranes of plant tissues, thus enhancing the extraction yield of the bioactive compounds from tomato residues while reducing the solvent consumption and the extraction time at once, has been proposed. Their utilization to complement traditional extraction methods can represent a suitable, effective, and environmentally friendly approach for the valorization of tomato by-products [22]. 

Indeed, the increasing consumers’ interest in chemical-free products has created motivation for producers to apply innovative technologies for the recovery of natural compounds to be used in the food, cosmetic, and pharmaceutical industries [4]. High-pressure homogenization (HPH), pulsed electric fields (PEF), supercritical fluid extraction (SFE-CO_2_), ultrasound-assisted extraction (UAE), and microwave-assisted extraction (MAE) have been demonstrated to effectively enhance the extraction yields of bioactive compounds from a wide variety of agri-food by-products, including those from tomato processing [11,14,15,17,23].

Previous studies mostly focused on the application of the bioactive compounds derived from tomato processing by-products in the food industry, with a specific focus on lycopene [10], as functional ingredients for different products, such as baked goods, meat, and sauces [10,24]. This paper aims to discuss the recent advancements in tomato processing by-product valorization for the recovery of different valuable compounds, namely lycopene, polyphenols, cutin, pectin, oil, and proteins, focusing on the major conventional methodologies and the application of emerging technologies as energy efficient, and environmentally friendly methods to increase the effectiveness of bioactives recovery. To follow the near-zero discharge concept, the utilization of cascade extraction processes and biogas generation from tomato residual biomass have also been discussed.

## 2. Chemical Composition and Characteristics of Tomato Pomace, Peels, and Seeds

Tomato pomace generally consists of 56% pulp and peels and 44% seeds on a dry basis [25]. However, the composition of tomato pomace is simultaneously dependent on the type of final product and the peeling methods applied in the production line. The tomato pomace generated during the peeling phase of peeled tomato production only consists of peels without seeds. However, in the case of tomato juice and paste production, tomato pomace is a mixture of peels, seeds, and a small amount of pulp [26]. 

According to the literature, tomato pomace contains several high-added-value compounds, particularly lycopene, proteins, oil, and dietary fibers. More specifically, fibers are the main component of tomato pomace (39.11–59.03% on a dry basis), and other compounds, such as oil and proteins, range between 2–16.24% and 15.08–24.67%, respectively, depending on the variety, geographical location of the cultivation areas, growing stages, ripening, type of processing, extraction conditions, as well as analytical techniques utilized [24,25]. 

However, due to differences in the chemical composition of tomato peels and seeds, the main valuable compounds that can be recovered are largely different, as summarized in Table 1. 

In particular, tomato peels are a rich source of lycopene, representing 80–90% of the total carotenoids, polyphenols, and dietary fibers, as reported in Table 2 [31]. Tomato seeds are mostly composed of oil, containing 80% unsaturated fatty acids, mainly linoleic (48.2–56.1%), oleic (22.2–23.8%), and palmitic (12.3–17.2%) acids [35,36,38,39], and proteins, with glutamic acid (19.44–24.37%) and aspartic acid (8.82–10.32%) being the most abundant amino acids in tomato seed oil [40].

Therefore, tomato pomace has a great potential use as a rich and low-cost source of diverse high-added-value compounds that, due to their different natures and characteristics, have been demonstrated to possess antioxidant, anti-inflammatory, anti-aging, and anti-cancer properties, retard degenerative diseases, and control cholesterol and blood sugar levels [41]. 

The most recovered target compounds from tomato processing by-products and their potential diversified applications in several industrial sectors are summarized in Figure 2. 

## 3. Separation of Peels and Seeds from Tomato Pomace

As discussed in the previous section, tomato peels and seeds have some differences in their chemical composition, consequently resulting in different valorization procedures and the production of different compounds with wide potential utilization in different industrial sectors. In this respect, the first necessary step to effectively valorize by-products, particularly peels and seeds, is the separation of these two fractions [42]. 

Two possible routes can be used to separate tomato peels and seeds, constituting the pomace, namely wet and dry separation. Wet separation is based on the difference in density between the tomato peels and seeds. The pomace is mixed with water in a mixer-settler where seeds sink to the bottom, while peels, having a lower density, float at the top [40]. Kaur et al. (2005) designed a flotation system for tomato pomace componentization characterized by separation efficiencies of 69.17% and 48.29% for peels and seeds, respectively [43]. Shao et al. (2013) applied the wet separation method for tomato pomace and stated that by repeating the separation several times by reprocessing the obtained fractions, the purity of the tomato peels and seeds separation can be improved up to 90% (89.65% and 96.6% for peels and seeds, respectively), although this method caused significant loss of micronutrients [28]. 

The dry separation method consists of a drying step for tomato pomace, which is then fed onto a cyclone by an air flow where the separation of the two fractions takes place. The peels move upward, exiting the cyclone from the upper outlet section with air, while the seeds, which are heavier, move downward in the opposite direction of air flow and leave the cyclone from the bottom outlet section. Shao et al. (2015) optimized the separation step of tomato pomace using an air aspirator system using response surface methodology (RSM). They reported that tomato pomace with a moisture content of 8% had a separation efficiency of 68.56% when the air velocity and feed rate were 6.4 m/s and 40 kg/h, respectively [42]. 

Wet and dry separation methods have advantages and disadvantages. Wet separation is a water-consuming process and is suitable for fresh pomace to obtain seeds with high purity. In contrast, dry separation is an energy-consuming process and allows acquiring peels with high purity (mostly due to the lower yield). However, although the dry separation is energy intensive, it is superior in preserving valuable water-soluble compounds present in peels and seeds with respect to the wet counterpart, which causes significant losses of micronutrients. In addition, from an environmental perspective, wet separation generates wastewater and consequently water pollution, while dry separation results in air pollution and dust, which can lead to health issues among workers. Overall, the selection of the two alternative separation methods for the peels and seeds from tomato pomace should be closely linked to the requirements of the subsequent extraction stage, the eventual pre-treatment processes of the biomass, and the technologies applied for the recovery of the compounds of interest [24,28]. A schematization of the wet and dry methods for tomato peel and seed separation has been presented in Figure 3.

## 4. Recovery of High-Added-Value Compounds from Tomato Processing By-Products by Conventional Methods

### 4.1. Lycopene Characteristics, Applications and Extraction

Lycopene is a lipophilic antioxidant belonging to the carotenoid family, synthesized by many plants and essential for light absorption during photosynthesis and protection against photo-oxidative damage. It is a bright red carotenoid found in red fruits and vegetables and represents more than 85% of the total carotenoids in tomatoes [44]. 

For many years, lycopene was commonly used as a pigment and natural food colorant [4], highly accepted by consumers as a food additive due to its well-known antioxidant properties and consequent health benefits, such as the reduction of the risk of coronary heart disease and atherosclerosis. Likewise, epidemiological findings have linked a lower risk of the incidence of specific types of cancer with the consumption of lycopene [44,45]. Moreover, in vivo experiments have proven that lycopene exerts considerable biological effects, including immunomodulatory and anticancer activity against prostate cancer, breast cancer, cardiovascular illness, and neurological degenerative diseases [46]. Lycopene has been categorized as a class A nutrient by the World Health Organization (WHO) and the Food and Agriculture Organization of the United Nations (FAO) based on its diverse applications in the food, medicine, and cosmetics industries [47].

Since lycopene is not synthesized in the human body, industrial lycopene production from tomatoes is highly demanded by food and pharmaceutical companies to produce functional foods and nutraceuticals. 

In fact, according to the report of Global Opportunity Analysis and Industry Forecast, the lycopene global production in 2020 generated USD 107.2 million, and it is predicted to make USD 187.3 million by 2030, witnessing a compound annual growth rate (CAGR) of 5.2% from 2021 to 2030 [45]. 

According to the data found in the literature, the highest concentration of lycopene, ranging from 72% to 90%, is found in tomato skins and water-insoluble parts of tomatoes and is five times higher than that found in tomato pulp [48]. Therefore, lycopene recovery from tomato processing by-products has great potential interest [11]. 

Lycopene is a non-polar compound commonly extracted from tomato peels using organic solvents. In addition to the individuation of the most appropriate solvent, the optimization of effective processing parameters, including temperature, solvent to sample ratio, mixtures of different solvents, and extraction time, is necessary to improve the extractability of target compounds [49,50].

According to most of the studies present in the literature, the traditional way to recover lycopene from tomato peels is through the use of solvent mixtures that have been proven to enhance the extraction yield [49,51]. Zuorro (2020) utilized optimized mixed-polarity solvent mixtures, namely n-hexane–ethanol–acetone and ethyl acetate–ethanol–acetone, to extract lycopene from tomato peels and demonstrated that the highest lycopene extraction yield (95%) was obtained with a mixture containing 30.6% hexane, 32.8% ethanol, and 36.6% acetone (*w*/*w*) at 40 °C. This mixture was effective in producing a tomato oleoresin with high lycopene content (12.7 wt%) and antioxidant capacity (1582 µmol TE/g of tomato peels) [51]. 

The optimization of the solid-liquid extraction process to maximize the recovery of lycopene from tomato pomace was also investigated by Pandya et al. (2017) by selecting the most effective combination of solvents, extraction temperature, time, and solid-liquid ratio. The authors showed that using a mixture of acetone–ethyl acetate (1:1) as solvent at 40 °C for 5 h and a feed-to-solvent ratio (*w*/*v*) of 1:30 resulted in the highest yield of lycopene (6.11 mg/g of tomato pomace) [49]. The effects of these key parameters on the recovery of carotenoids from tomato by-products were also analyzed by Strati et al. (2011). The authors reported that, regardless of the type of solvent used, lycopene extraction was considerably influenced by the number of extraction steps. Among all the investigated solvents, ethyl lactate allowed obtaining the highest lycopene yield (0.243 mg /g dw) at 70 °C after 30 min of extraction. However, even at 25 °C, a similar extraction yield (0.202 mg/g dw) was obtained using ethyl lactate, suggesting that this solvent is capable of extracting at ambient temperature more lycopene than other solvents used at higher temperatures, thus reducing energy consumption and costs [52]. 

Although most studies focused on tomato peels and tomato pulp, another by-product of the tomato processing industry that contains a not-insignificant amount of lycopene, was investigated in several studies as a source of this carotenoid [48]. Poojary et al. (2015) utilized factorial design methodology for the optimization of extraction processing conditions to obtain high purity all-trans-lycopene from tomato pulp waste. In this work, the authors evaluated the effect of four variables, namely extraction time (1–60 min), temperature (30–50 °C), concentration of acetone in hexane (25–75%, *v*/*v*), and solvent amount (10–30 mL), on lycopene recovery yield. Experimental results showed that the highest lycopene amount (0.038 mg/g) was acquired at 30 °C for 60 min by using 30 mL of a mixture of 25% acetone in hexane (*v*/*v*) as solvent. However, numerical results demonstrated that the optimal extraction conditions were 20 °C for 40 min using 40 mL of solvent mixture, resulting in 0.039 mg/g lycopene in tomato pulp and limiting the isomerization and degradation of all-trans-lycopene. From an industrial point of view, these optimal extraction conditions are particularly appropriate, resulting in a high level of purity and recovery of all trans-lycopene (98.3% and 94.7%, respectively) [53]. Moreover, over the past few years, researchers have been motivated to use green solvents for lycopene extraction. In this regard, Amiri-Rigi and Abbasi (2019) evaluated the application of olive oil microemulsions as a green solvent by applying different proportions of olive oil, lecithin, 1-propanol, and water to examine their abilities to extract lycopene from tomato pomace. The results showed that the highest extraction efficiency (88%) was obtained when mixing 1 gr of tomato pomace with 5 gr of microemulsion prepared with the combination of olive oil: water: lecithin: 1-propanol (10:10:53.33:26.67 wt%). More importantly, this food-grade microemulsion enriched in lycopene can be used in a variety of formulations in the food industry because of its good solubility in aqueous and non-polar media and can enhance the health-promoting qualities of both lycopene and olive oil [54].

### 4.2. Cutin Characteristics, Applications and Extraction

Cutin is the main constituent of the plant cuticle (40–85%, *w*/*w*), which is the exterior layer protecting the epidermis of leaves, aerial parts of plants, and fruits. Cutin protects the epidermis from the environment, including humidity, UV radiation, temperature oscillations, pathogen attack, water loss, and gas exchange, and has attracted the interest of researchers to produce synthetic compounds mimicking its action to be applied in packaging materials, UV filters, and membranes [55]. Additionally, as cutin is biodegradable, it could represent a suitable alternative to oil-based polymers to reducing the environmental burdens due to their manufacturing, usage, and disposal [56].

The molecular structure of cutin is characterized by an amorphous branching with a flexible three-dimensional network mostly comprised of C_16_, C_18_, or a combination of both fatty acids that interacts with polysaccharides, waxes, and phenolic compounds [57]. These long-chain fatty acids (called cutin acids) can be considered novel building-block chemicals possessing specific reactive polyfunctional characteristics to be employed in the pharmaceutical industry [58], synthetizing innovative bio-resins and lacquers that can be applied as an appropriate internal protective coating to metal containers for food packaging [59]. Tomato processing by-products have been proposed as a renewable source for biopolymers due to their high content of cutin. In this regard, Cifarelli et al. (2016) investigated cutin extraction from tomato peels by various extraction procedures, including alkaline hydrolysis (method A), the use of sodium carboxylate (method B), and sodium hydroxide/hydrogen peroxide (NaOH/H_2_O_2_) (method C). Results revealed that, regardless of the method used, the principal component of tomato cutin, 10,16– dihydroxyhexadecanoic acid, was extracted with a yield ranging between 83% and 96%. Products obtained using both B and C methods showed lower polydispersity and higher purity than those obtained from alkaline hydrolysis. However, through alkaline hydrolysis, a gummy mass of cutin is acquired, representing the most interesting product that could be potentially used in bio-resin formulation [58]. 

In another study, Cicognini (2015) investigated cutin extraction from tomato peels involving stages of thermal treatment, filtration, acidification, centrifugation, and drying, resulting in an extraction yield of 25 ± 2% [60]. Interestingly, Manrich et al. (2017), by applying the same sequence of unit operations, produced hydrophobic edible films containing pectin acting as a binder. Their results showed that the films produced by cutin/pectin (50/50 *w*/*w*) presented a uniform structure in which cutin completely diffused into pectin. They stated that conformity tests (water uptake and solubility tests) proved the influence of lipophilic cutin on the lower water uptake and solubility of cutin/pectin film [61].

Since cutin consists of esterified C_16_ and C_18_ fatty acids with specific characteristics, such as biodegradability, UV-blocking, non-toxicity, insolubility, and impermeability [56], Benítez et al. (2018) focused on the valorization of fatty acids from tomato pomace to exploit these attributes and produce bio-based materials. The extraction procedure, consisting of an alkaline hydrolysis and thermal treatment followed by neutralization with HCl 3 N, allowed them to produce a combination of unsaturated and hydroxylated fatty acids. According to their results, the extraction conditions, namely NaOH 0.5 N, a temperature of 100 °C, and a processing time of 6 h, resulted in optimal conditions with a yield of ~31%, *w*/*w* [56], slightly higher than that obtained by Cicognini (2015) [60].

### 4.3. Pectin Characteristics, Applications and Extraction

Pectin is present in the primary and secondary plant cell walls and consists of a complex set of polysaccharides. It is a complex galacturonic acid (GalA)-rich polymer [62], which, according to the FAO, must contain at least 65% of GalA [63]. This compound is a biocompatible polysaccharide with high availability, low production cost, and inherent biological characteristics strictly depending on the type of source and extraction processes. 

It is the most used biopolymer worldwide, whose market is predicted to grow from USD 1 billion in 2019 to USD 1.5 billion in 2025 [64]. 

For many years, the food and beverage industry has successfully used pectin as a gelling agent, thickening medium, and stabilizer due to its physicochemical characteristics, such as hydrogel-forming capacity, which make it suitable for application in hydrated and viscous foods [65]. Currently, there is a growing interest in its application in the biotechnological and pharmaceutical industries due to its potential health benefits, such as anticancer activity and cholesterol reduction. It is also used for gene delivery, wound healing, and to produce artificial corneas and contact lenses [66,67]. More importantly, low-methoxy pectin can form gels with calcium ions (Ca^2+^), making this compound suitable to be used as a matrix for the entrapment and delivery of drugs, cells, and proteins [68].

In addition, pectin can be used in the production of packaging material, edible coatings, and as a tin corrosion inhibitor in metal packaging used in the canning industry. The latter potential utilization was suggested by Grassino et al. (2016). The authors investigated the extraction of pectin from dried tomato peels. After milling, peels were loaded under reflux in a condensation system at 90 °C, using ammonium oxalate and oxalic acid as extracting solvents, and pectin was recovered in two extraction steps, lasting 24 h and 12 h, respectively. The highest pectin yields in the first and second steps were 15% and 32.6%, respectively. However, the quality of the pectin extracted in the second step was lower due to the presence of methoxy and anhydrouronic acids and the increased esterification degree, suggesting that higher pectin yields are not necessarily associated with higher pectin quality. The authors evaluated the characteristics of the pectin extracted from tomato peels to propose its application in the canning industry. The results allowed us to conclude that this material has an inhibition capacity (73%) against corrosion in cans, higher than that of commercially available pectin (about 60%), probably due to the synergistic effects exerted by the several compounds present in tomato peels [32]. 

Likewise, Alancay et al. (2017) investigated the extraction of pectin from tomato by-products. The authors applied aqueous and acidic extraction methods and compared the results. The optimal processing conditions for acidic extraction with hydrochloric acid solution at a pH of 2 were 85 °C and 60 min, while distilled water at 95 °C was used for aqueous extraction, and the optimal extraction time was 180 min. Results demonstrated that the characteristics of the pectin obtained by applying the acidic method, e.g., high purity and gelling power, were similar to those of the commercial pectin, which makes it usable in food formulation [69]. The extraction with acids was also proven effective by Morales-Contreras et al. (2017). The authors concluded that an extraction yield of 19.8% of pectin can be obtained from tomato husk waste utilizing acidification with 0.1 N HCl, a thermal treatment at 100 °C for 20 min, followed by a precipitation step with ethanol 95% (*v*/*v*) [70].

### 4.4. Oil Characteristics, Applications and Extraction

The physicochemical characteristics of vegetable oils are important parameters to assess their acceptability by consumers, who are increasingly interested in purchasing oils with lower health risks and higher health benefits than animal fats [37]. Tomato seeds can be regarded as a great source of vegetable oil (17.8–24.5 g/100 g seed) rich in bioactive compounds, such as polyphenols, tocopherols, and phytosterols, with good antioxidant capacity [36] and beneficial properties for human health [24], including improving the immune system and blood pressure, as well as preventing aging and arteriosclerosis [71].

Beyond these bioactive compounds, tomato seed oil predominantly contains saturated and unsaturated fatty acids up to 14–18% wt and 76–80% wt, respectively [35,38,39], being stearic (C18:0, 5.2–5.4%), palmitic (C16:0, 12.3–17.2%), oleic (C18:1, 22.2–23.8%), linoleic (C18:2, 48.2–56.1%), and linolenic (C18:3, 2.1–2.7%), the major fatty acids [72]. In particular, among the polyunsaturated fatty acids (PUFAs), a group of fatty acids, namely linoleic acid and alpha-linolenic acids, exerting many physiological functions, such as controlling blood pressure or cell signaling, are not synthesized by the human body [73]. Tomato seed oil, thus, represents a great source of PUFAs, and its recovery from tomato seed can be proposed to obtain a final product suitable for food, pharmaceutical, and cosmetic applications [72].

To maximize the extractability of oil from tomato seeds, an optimization step of the operating conditions involved in the extraction process is needed. Shao et al. (2012) proposed hexane as a solvent to extract oil from tomato seeds with high yield and antioxidant activity. The effect of processing conditions, including temperature, time, solvent/solid ratio, and particle size, was evaluated. The results demonstrated that the extraction yield was enhanced by increasing the extraction time, temperature, and solvent-to-solid ratio, while the oil extraction yield decreased with increasing the particle size. The maximum oil yield of 20.32% was obtained at the optimal extraction conditions, namely 25 °C, 8 min, a solvent/solid ratio of 5/1 (*v*/*w*), and a particle size of 0.38 mm [74].

With the aim of comparing different methods for oil extraction, Ozyurt et al. (2021) investigated conventional extraction (Soxhlet extraction), cold press extraction (CPE), and enzymatic-assisted aqueous extraction (EAAE). The amounts of oil extracted from tomato seeds were 13.07 ± 2.24%, 12.80 ± 0.13%, and 9.66 ± 0.50% using Soxhlet extraction, CPE, and EAAE, respectively. Moreover, the authors reported that the oil yield was 97.93% for CPE and 73.91% for EAAE, illustrating that conventional solvent extraction with hexane resulted in a higher amount of oil in comparison to the other two alternatives [75]. 

Plant oil quality and its antioxidative characteristics can be assessed by determining their tocopherol content. Indeed, tocopherols are among the most significant antioxidants found in vegetable oil, responsible for its stability and nutritional value [76]. In this regard, Botineştean et al. (2013) investigated the oil extraction from tomato seeds with different organic solvents (hexane, petroleum ether, and diethyl ether) to evaluate their effects on the tocopherol content in tomato seed oil. The authors reported that the highest tocopherol content in tomato seed oil was obtained by utilizing diethyl ether as an extracting solvent (115.5 mg/100 g). As predicted, since the molecular structure of a tocopherol includes a free hydroxyl group, its content in tomato seed oil increases with increasing the solvent polarity (the polarity index of diethyl ether is 27 times higher than that of hexane and petroleum ether) [76]. 

### 4.5. Proteins Characteristics, Applications and Extraction

Many research studies have shown that tomato seeds contain considerable amounts of nutrients and proteins account for approximately 20% to 40% of the total weight (on a dry basis) [19,29,35,36,37]. The major amino acids in tomato seeds are glutamic acid and aspartic acid [19,77], which are particularly suitable to be used as flavor enhancers in foods with umami and sour tastes [78].

According to the literature, the most abundant essential amino acids in tomato seeds are arginine, threonine, lysine, and leucine [72,77]. The high content of lysine (1.34%) in tomato seeds makes the proteins extracted from this source an interesting ingredient, particularly to improve the quality of proteins in cereal-based products and to produce protein-based food supplements with significant functional characteristics [78], playing a major role in calcium absorption, building muscle protein, and the production of hormones, enzymes, and antibodies [77,79]. 

Generally, the protein extraction process from tomato seeds consists of two stages, namely alkaline separation and isoelectric precipitation [19,40,78]. Meshkani et al. (2016) investigated the protein isolation from defatted tomato pomace (peels and seeds) and tomato seeds, whose protein contents were 35.29% and 44.65%, respectively. The optimization of the extraction process in terms of temperature (10–50 °C), alkaline and acidic pH (10–12 and 3.1–4.3), time (30–70 min), and solvent/solid ratio (1:10–1:50 *w*/*v*) was conducted to maximize protein recovery. Results demonstrated that the best extraction conditions were 37.73 °C as the extraction temperature, pH 12.00 for alkaline conditions, and pH 3.73 for acidic conditions, a solvent/solid ratio equal to 1:40, and 60 min as the extraction time. In these conditions, protein extraction yields of 86.84% and 64.15% were obtained from defatted tomato pomace and defatted tomato seeds, respectively [40]. 

Mechmeche et al. (2017) proposed the use of RSM to optimize protein extraction from defatted tomato seeds. The authors demonstrated that the optimal extraction conditions were 82.81/1 (*v*/*w*) water/solid ratio, 49.76 h of extraction time, and 24.56 min of mixing time, with a constant pH of the suspension equal to 7.5 during the extraction. Confirmatory studies carried out under these conditions resulted in an 80.37% protein yield, which was completely in agreement with the value predicted by the model (81.22%) [19]. 

Generally, in tomato processing, thermal treatments, such as hot break (85–100 °C) and cold break (60–65 °C or ambient temperature), are used to completely or partially inactivate pectolytic enzymes and increase the consistency and viscosity of the product [78,80]. Shao et al. (2014) investigated the effects of the two abovementioned thermal processes on the isolation and functionality of proteins from tomato seed meal. According to their results, hot break resulted in a protein extraction yield from defatted tomato seeds in the range 9.1–26.3%, while cold break determined a protein extraction yield from defatted tomato seeds in the range 25.6–32.6%. The lower temperatures used in cold break led to these findings [78]. The thermal treatments of fruits during tomato processing affect waste valorization, as confirmed by Szabo et al. (2021). The authors have demonstrated that industrial thermal processes can adversely affect the bioactive compounds contained in tomato seeds and peels, including proteins [36].

## 5. Application of Green Technologies for Tomato Processing By-Product Valorization

Conventional solvent extraction (CSE) is among the most widely used commercial methods to recover high-added-value compounds, such as carotenoids, fibers, oil, and proteins, from tomato processing by-products. However, some disadvantages are associated with this traditional unit operation, particularly the high extraction time and temperature requirements and the use of organic hazardous solvents, which negatively affect the environmental sustainability of the process and the safety of the extracted compounds [81]. Therefore, the application of novel technologies enabling to weaken or disrupt the cell membrane of the plant tissue to unlock the bioactive compounds can represent an effective strategy to mitigate the limitations associated with CSE and increase the recovery of high value-added compounds from tomato residues [82]. The main advantages deriving from the implementation of different emerging techniques, such as high-pressure homogenization (HPH), pulsed electric fields (PEF), ultrasounds (US), supercritical fluid extraction (SFE-CO_2_), and microwaves (MW), in the extraction process and the comparison of their effectiveness to boost the recovery of bioactives from tomato processing by-products have been comprehensively summarized in Table 3 and are better discussed throughout this paper. 

Although these technologies present several advantages with respect to CSE, their utilization to complement tomato by-product valorization processes also shows disadvantages, such as high investment costs and high energy consumption, as reported in Table 4, where the main features and the associated benefits are also highlighted.

### 5.1. High Pressure Homogenization (HPH) Technology

In HPH technology, one of the most effective mechanical methods for large-scale cell disruption, a fluid containing suspended solids is pumped through a tight gap valve utilizing a high-pressure intensifier, followed by depressurization with the subsequent generation of high shear and elongational stresses, and cavitation. Consequently, cells, particles, or macromolecules suspended in the fluid are exposed to a high level of mechanical stress, getting deformed and twisted, as shown in Figure 4 [99]. Many studies have evaluated the use of HPH for microbial inactivation in the food industry. However, this technology has been effectively applied as a method to improve the release of intracellular bioactive compounds from agri-food by-products as a result of the complete disruption of the plant cells induced by HPH [15]. Since HPH is considered a physical treatment and needs no or a little amount of organic solvent, it is considered a very environmentally friendly method to recover target compounds from agro-industrial biomass [15]. However, it is an intrinsically non-selective operation with high energy costs associated and requiring high capital investments [15,100].

In 2019, Jurić et al. applied HPH technology (1–10 passes, at 100 MPa) as a disruption technique to recover bioactive compounds from tomato peels by using only water as a solvent. The results showed that HPH reduced the size of tomato peel suspensions, leading to the complete disruption of single plant cells that consequently released high-added-value compounds. Particularly, when the number of HPH passes through the valve increased, the cell disruption occurred completely, and consequently, greater quantities of total polyphenols and proteins were released. The authors also reported that in comparison with high-shear mixing (5 min at 20,000 rpm), referred to as the control, HPH processing (10 passes) led to higher release of intracellular compounds, including polyphenols (+32.2%) and proteins (+70.5%) and a rise in antioxidant activity (+23.3%). Interestingly, in terms of lycopene yield, the authors compared HPH with other methods, including CSE [85], SFE-CO_2_ [96], PEF-assisted extraction [85], and UAE [87]. The results showed that HPH led to the highest lycopene yield among all the compared methods. The quantitative results were also confirmed by the HPLC analyses, confirming that the amount of lycopene recovered from tomato peels (19.3 mg/g dw) was considerably greater than the values reported in the literature (from 0.5–0.8 mg/g dw) [31] and reached the maximum amount of 1.5 mg/g dw [107]. More importantly, its recovery was performed via a sustainable, green, and entirely physical method, and the final products could be applied in the formulation of functional foods or mixed with the peeled tomato products to enhance their bioactivity [15].

Since HPH is a physical treatment that results in the complete disruption of plant cells, it can be very effective in recovering high-molecular-weight compounds from plant cell tissues, such as pectin. Van Audenhove et al. (2021) investigated the application of the HPH-facilitated acid extraction method to recover pectin from tomato processing residues. In this study, an industrial method for pectin recovery was applied by carrying out the extraction process in a single step with nitric acid (pH approximately 1.6). Additionally, HPH technology was tested to facilitate a further step for the recovery of the pectin fraction, which remained unextracted after the treatment with nitric acid. The results showed that polysaccharide cell walls were considerably affected by HPH (20 MPa for a single pass), and nearly two thirds of the residual pectin was recovered in the subsequent extraction step. Overall, the results of this study demonstrated the potential of HPH technology to enhance pectin extraction from tomato processing residues compared to the conventional acid extraction [100].

### 5.2. Pulsed Electric Fields (PEF) Technology

PEF is a non-thermal electrotechnology that involves the exposure of plant tissue suspensions, placed between two metal electrodes, to repetitive short duration pulses (1 μ^−1^ ms) of moderate electric field (0.5–10 kV/cm) and relatively low energy input (1–20 kJ/kg), leading to the permeabilization of cell membranes by pores formation [108]. Being electroporation the outcome of PEF pre-treatment, mass transfer of the intracellular compounds during the subsequent extraction step is enhanced, whereas the solvent consumption, the extraction time, and the energy costs are decreased [101,102,103]. The schematic of a continuous PEF treatment chamber is depicted in Figure 5.

Andreou et al. (2020) applied PEF technology in three different stages of industrial tomato processing, including peeling, juice extraction, and tomato waste valorization. The recovery of high-added-value compounds from residues of the juicing process assisted by PEF was investigated. As a result, the application of PEF at 2 kV/cm and 700 pulses enhanced the extraction yield of carotenoids by 56.4% and doubled the concentration of total phenolics compared to the untreated samples [83].

With the aim of decreasing the required thermal energy in the thermophysical peeling stage, PEF technology can be implemented in the washing stage of tomato processing, with the consequent extent of improving lycopene extraction from tomato peels [83,109]. Pataro et al. (2018) demonstrated that PEF pretreatment enabled the permeabilization of the plant cells and facilitated the detachment of the peels from the fruits in the following steam blanching (SB) and peeling steps. The application of PEF contributed, at the same time, to the promotion of an increase in the recovery yield of bioactive compounds from tomato peels. The combination of PEF (0.5 kV/cm, 1 kJ/kg) and SB (at 60 °C for 1 min) synergistically enhanced carotenoids extraction, resulting in 37.9 mg/100 g of fresh tomato peels, 1.7 times higher than that obtained by applying only SB [17].

In addition, the same authors evaluated the influence of PEF pre-treatment and extraction temperature on the recovery of carotenoids from tomato peels in 2019. Different PEF treatment conditions were applied with field strengths, E, in the range 0.5–5 kV/cm and energy input, W_T_, in the range 0.5–20 kJ/kg, and the effects of different temperatures (20 to 50 °C) on lycopene extraction yield were assessed by using acetone as solvent. The optimal PEF processing conditions were E = 5 kV/cm and W_T_ = 5 kJ/kg, which resulted in an extraction yield of total carotenoids that was 47% higher than that obtained from untreated peels, and the antioxidant power of the extract was 68% higher than that of the untreated tomato peels. Moreover, regardless of the PEF treatment, increasing the extraction temperature from 20 to 50 °C increased the lycopene extraction yield by about 22% [84].

Industrial tomato by-products provided by a canning company were used by Pataro et al. (2020) to investigate the effect of PEF pre-treatment on the recovery yield of lycopene from industrial tomato peels. PEF processing was carried out at different field strength (E= 1–5 kV/cm) and energy input (W_T_ = 5–10 kJ/kg), and ethyl lactate and acetone were used as solvents. In line with the study previously discussed, the results demonstrated that the application of PEF pre-treatment (5 kV/cm, 5 kJ/kg) considerably increased the lycopene extraction rate (27–37%), the antioxidant power of the extract (18.0–18.2%), and the lycopene recovery yield (12–18%) with respect to the untreated samples. Moreover, acetone allowed to extract a higher amount of lycopene (17.5 mg/g dw) with respect to ethyl lactate (10.14 mg/g dw), indicating a higher ability of this solvent to penetrate into the plant cells, enabling a higher amount of intracellular lipophilic compounds to dissolve in it [11]. The application of PEF technology for carotenoid extraction from tomato wastes was also investigated by Luengo et al. (2014). The authors applied PEF pre-treatment of different intensities (3–7 kV/cm and 0–300 µs) and used a mixture of hexane: acetone: ethanol (50:25:25) as solvent, with the aim of improving extraction yield and reducing the total amount of solvent in the extraction step. PEF pre-treatment (at 5 kV/cm and 90 µs) on tomato peels increased carotenoid extraction by 39% compared to the untreated samples. More importantly, RSM results showed that the application of PEF reduced the hexane consumption from 45 to 30% at the same extraction yield [85].

### 5.3. Ultrasound Technology

Ultrasound-assisted extraction (UAE) is an alternative process to CSE that is able to induce cell wall disruption, principally attributed to acoustic cavitation. Cavitation is produced by the interaction between the liquid, the ultrasonic waves, and the gas dissolved in the liquid. As presented in Figure 6, UAE creates cavitation bubbles using high frequency pulses, and local hotspots at the macroscopic scale with high shear stress and temperature. The penetration of the solvent into the plant cells is favored, as well as the release of intracellular compounds [105]. Low-frequency ultrasound (16–100 kHz) can be applied for the extraction of valuable compounds such as hydrophobic carotenoids (lycopene, beta-carotene, capsaicin, and lutein) and hydrophilic flavonoids (anthocyanins, tannins) from agricultural by-products [110]. In general, the application of relatively mild ultrasonic conditions stimulates the release of the compounds entrapped in the intracellular and extracellular spaces of the plant tissues, allows increasing the mass transfer rate and decreasing the extraction temperature and time, and, consequently, improves the extraction efficiency. However, US is a non-selective extraction method, and its application could damage thermolabile compounds due to the local increase in temperature [22,91,92,98].

Kumcuoglu et al. (2013) investigated the application of ultrasound technology for lycopene extraction from tomato paste processing by-products. The authors compared the recovery yield of lycopene obtained with UAE and conventional organic solvent extraction (COSE) [87], using as solvent a mixture of BHT (butylated hydroxytoluene) 0.05% (*w*/*v*), hexane, acetone and ethanol (2:1:1). The maximum extraction yield was obtained by applying ultrasounds at a power of 90 W, a liquid/solid ratio of 35:1 (*v*/*w*), and an extraction time of 30 min, while the processing conditions for COSE were a liquid/solid ratio of 50:1 (*v*/*w*), a temperature of 60 °C, and an extraction time of 40 min. Results revealed that UAE could extract about 80% of lycopene after 10 min of treatment, whereas the extraction time was at least 20 min using COSE. Overall, the comparison between COSE and UAE revealed that UAE is a more efficient method that requires a lower amount of solvent (a reduction of about 30%) and a shorter extraction time than COSE [87].

Silva et al. (2019) investigated the application of UAE for the extraction of lycopene from tomato wastes using a mixture of eco-friendly solvents, namely ethyl lactate and ethyl acetate, to increase the sustainability of the process. The highest amount of lycopene (1.33 mg/g dw) was obtained in the optimized UAE processing conditions. The extraction yield of lycopene obtained utilizing the same solvents without sonication was 1.209 mg/g dw, thus 9.4% lower than that obtained upon UAE. By coupling less harsh solvents, such as ethyl lactate and ethyl acetate, with ultrasound technology, the lycopene extraction yield was enhanced, this representing a greener approach for the lycopene recovery in comparison to organic solvents [88]. In 2019, the same authors studied a sustainable method for lycopene extraction from tomato processing waste by applying UAE and hydrophobic eutectic solvents (HEMs), namely lactic acid as hydrogen-bond donor (HBD), and DL-menthol as hydrogen-bond acceptor (HBA). In this study, the extraction conditions that maximized the lycopene content in the extract (1.45 mg/g dw), as determined by RSM, were 120 mL/g solvent-solid ratio, 70 °C, and 10 min [50].

In order to improve the extraction of lycopene from tomato waste, the combination of sonication and edible solvents (sunflower oil) was exploited as a green extraction strategy by Rahimi et al. (2019). In this study, response surface methodology (RSM) was used to select the best experimental conditions, including the solid to oil ratio (S/O), ultrasonic intensity (W/m^2^), and extraction time (min), that maximize the lycopene yield. Results showed that, while the ratio of solid to oil had a slight impact on the yield of extraction, the time and ultrasonic intensity significantly affected the total lycopene recovery. The maximum lycopene yield (81.57%) was achieved using an ultrasonic power of 70 W/m^2^, a solid-to-liquid ratio of 20 (*v*/*w*), and an extraction time of 10 min. The proposed approach complies with the concept of green processes since it permits the use of renewable resources, ensures product safety and quality, and allows to obtain pigmented oils that can be used as sources of lycopene in a variety of products [111].

Similarly, Pettinato et al. (2022) focused on the UAE optimization using ethanol as solvent, analyzing the effects of the different variables involved on the lycopene extraction yield from tomato waste. The optimal extraction conditions were 65 °C, 20 min, liquid-solid ratio of 72 mL/g, US amplitude of 65%, and 33 s for pulse duration, resulting in a lycopene yield of 1.54 ± 0.05 mg/g dw [89]. However, a slightly lower temperature and a longer extraction time were required to obtain lycopene extraction yields comparable to those reported by Silva et al. (2019).

Said et al. (2020) investigated the lycopene extraction from both lab-prepared and industrial tomato wastes by using US (45 min at 50 Hz) and freeze drying. US, freeze-drying, and the combination of the two processes resulted in a recovery of 45.51 ± 1.84, 104.10 ± 1.23, and 138.82 ± 6.64 µg lycopene/g fresh tomato waste, with an increase of the lycopene extraction yield from industrial tomato waste of 0.8, 2.8, and 4.12 folds, respectively [112].

Sonication can also be applied for pectin recovery from tomato by-products. It has been demonstrated that US intensifies the extraction process, avoiding the long extraction times (12–24 h) required with the most common conventional methods [91]. Grissino et al. (2016) explored the possibility of using UAE (at 37 kHz) and CSE (by ammonium oxalate/oxalic acid in two steps) for the recovery of pectin from tomato residues. The highest pectin yields in the first step were 18.5% and 21.1% for UAE and CSE, respectively. However, for rich similar pectin extraction yields, CSE required 1440 min, which is two orders of magnitude higher than that required for UAE (15 min). Moreover, the second extraction step was ensuring higher pectin yields compared to the first step not only for CSE (31.2%), but also for UAE (36%). Therefore, the authors concluded that the main advantage of UAE was to considerably shorten the extraction time, making the process more environmentally friendly [86]. In another study, Singh Sengar et al. (2020) studied pectin extraction from tomato peels by using five different extraction methods, namely UAE, MAE, ohmic heating-assisted extraction (OHAE), ultrasound-assisted microwave extraction (UAME), and ultrasound-assisted ohmic heating extraction (UAOHE) at different power levels. According to their results, the yield of extracted pectin ranged from 9.30% for OHAE to 25.42% for MAE. They reported that although MAE led to a higher yield, UAME can be regarded as a greener extraction method with respect to the other extraction processes tested, allowing for comparable extraction yield as well as higher pectin quality [91].

US technology has also been proposed to increase the oil extraction yield from tomato seeds. Aarabi Arabani et al. (2015) applied sonication as a pre-treatment stage of tomato seeds to extract oil. Results demonstrated that the particle surface bonds can be weakened by combining US pre-treatment with other physical techniques, effectively enhancing the extraction yield. Immersion of tomato seeds in water for 24 h at 40 °C, followed by grinding and sonication (at 550 W and 37 kHz for 90 min), resulted in about 28.11% yield in oil extraction from tomato seeds. In fact, soaking the seeds in water before milling provided the needed texture of the lignocellulosic samples and prevented them from absorbing nonpolar and hydrophobic solvents or effusing the oil. Additionally, the milling process led to better penetration of the solvent into the sample. Overall, using combined pre-treatment processes (Hot water + Grinding + US) increased the oil extraction yield by 15.91% with respect to that obtained from untreated samples [92].

### 5.4. Supercritical Fluid Extraction

Supercritical fluid extraction (SFE) is an environmentally friendly method operated at the pilot and industrial scales. The schematic of the process is shown in Figure 7. This method is based on the increased solvating power of gases beyond their critical point [113]. Currently, one of the most frequently used supercritical fluids is carbon dioxide due to its beneficial characteristics, including low critical temperature and pressure, high purity, and low cost. More importantly, carbon dioxide can be applied for the extraction of compounds that are thermally unstable and cannot be purified by steam distillation. Since lycopene is a high-added-value bioactive compound and degrades easily when subjected to thermal processing, supercritical carbon dioxide-assisted extraction (SFE-CO_2_) can be applied as a green extraction process to recover tomato oleoresin rich in lycopene. Moreover, the selectivity of the extraction process can be increased by adjusting and monitoring the pressure and temperature of the system [114,115].

Mihalcea et al. (2021) studied the application of SFE-CO_2_ to tomato peels for lycopene-enriched oleoresin extraction and its microencapsulation to produce high-added-value ingredients with several potential applications. The optimization step of the processing conditions showed that SFE-CO_2_ carried out for 155 min at a pressure of 400 bar and a temperature of 74 °C resulted in the recovery of oleoresin form tomato peels with the highest lycopene content (5.28 mg/g dw) [94]. To evaluate the effect of SFE-CO_2_ on the extractability of intracellular bioactive compounds from industrial tomato residues, also Kehili et al. (2017) investigated the recovery of lycopene and β-carotene from tomato peels obtained in tomato industrial processing. The results showed that SFE conducted at the optimal operating conditions, namely 400 bars, 80 °C, 105 min, 4 g CO_2_/min, and 0.4 g CO_2_/g peels, led to a maximum lycopene recovery of 0.73 ± 0.03 mg/g dw. Moreover, these results were compared to those obtained by applying the conventional extraction methods (overnight at 200 rpm and 25 °C) using ethanol, ethyl acetate, and hexane as solvents. Interestingly, the SFE-CO_2_ extraction technique resulted in a higher yield of lycopene in comparison with conventional methods (+156%, +128%, +20%, by using ethanol, ethyl acetate, and hexane, respectively) [93]. Additionally, Hatami et al. (2019) evaluated, both experimentally and by applying mathematical modeling, the lycopene recovery yield by SFE-CO_2_ from tomato pomace, demonstrating that the greatest effect on lycopene recovery is related to the peel/seed ratio, followed by pressure, and temperature [96]. Considerably, the combination of pressure–temperature showed a positive synergistic effect on the lycopene recovery. Pressure affected lycopene recovery more effectively at higher temperature. Due to the higher availability of lycopene in the peels compared to the seeds, the optimized experimental conditions (80 °C, 500 bar, and a peel/seed ratio of 70/30) resulted in the highest lycopene recovery (0.358 mg of extract/kg of raw material) [96]. The effect of these parameters on lycopene extractability from tomato peels under SFE-CO_2_ was also investigated by Pellicano et al. (2020). The authors stated that applying SFE-CO_2_ at 550 bar for 80 min resulted in the highest oil extraction yield (79%), being the oil rich in lycopene (8.6 mg/kg dw) and β-carotene (15 mg/kg dw) [116].

Adapting to the zero-emission process concept, the application of SFE-CO_2_ technology for the extraction of high-quality oil from tomato processing by-products (seeds and peels) was also investigated by Lisichkov et al. (2011). In this study, the authors evaluated the effects of different parameters on the oil extraction yield. The maximum solubility of oil was obtained at operating pressures ranging from 210 bar to 280 bar at 40 °C for 2.5 h, which resulted in 0.25 g/g tomato seeds and peels with a solubility of 14 mg/dm^3^s [95].

### 5.5. Microwave Technology

Microwave technology consists of an indirect way of heating materials by means of electromagnetic radiation at wavelengths comprised between ordinary radio waves, and infrared radiation in the frequency range between 300 MHz and 300 GHz, although in most common applications the frequency range is comprised between 1–40 GHz. This process leads to the evaporation of moisture inside the plant cells, which in turn causes an increase in the pressure exerted on the cell wall. Subsequently, modifications of the physical and biological characteristics of the vegetable tissue occur, leading to improved penetration of the extracting solvent into the biomass and increased extraction yields of the target intracellular compounds [82]. A schematic of MAE is reported in Figure 8. A large variety of factors can affect MAE, such as frequency, time and microwave power, moisture content and size of the sample, solvent type and concentration, solid/liquid ratio, extraction time as well as the number of extraction cycles [105]. The MAE process has been shown to be an environmentally friendly technology that enables greater extraction yields in shorter time and energy usage than traditional processes, and that can even be carried out without the use of solvents. However, this technology also has some drawbacks that need to be monitored, such as poor selectivity, non-uniform heating, and limited penetration of the microwaves, which could lead to reduced extraction efficiencies. In addition, thermal degradation of phenolic compounds and changes in the chemical structure of the target compounds due to overheating might occur, causing undesired effects on their bioactivity and hindering their usability [82,98,105,106].

Lycopene and beta-carotene, the main polyphenols in tomato wastes, are very thermosensitive compounds that can be detrimentally affected by the extraction conditions. Lasunon et al. (2021) evaluated the effectiveness of MAE at different processing conditions to extract bioactive compounds from tomato processing by-products. According to their results, the higher the microwave power and the extraction time, the greater the degradation of bioactive compounds would occur. Furthermore, bioactive compounds extracted under MAE processing conditions, which allowed the highest recovery yield, showed low antioxidant activity, indicating that degradation phenomena were likely to occur. Moreover, the overall performance indicator showed that at the best MAE conditions, namely 300 W applied for 60 s at a temperature not exceeding 77 °C, the bioactive compounds were recovered with a high yield and characterized by a high quality (lycopene: 5.74 mg/100 g dw and beta-carotene: 4.83 mg/100 g dw) [117].

With the aim of assessing the effect of MAE processing parameters, such as temperature, type of solvent, and time, on total flavonoids (TF), total phenols (TP), and phenolic compound recovery, Tranfić Bakić et al. (2019) utilized this technology for tomato peel valorization [97]. The authors demonstrated that the extraction time had a limited effect (*p* > 0.05) on TF, TP, and phenolic compound extraction, while temperature and type of solvent significantly affected the polyphenols extraction yield. According to their results, the average TP content was 53.12 g/kg, and the highest recovery yields were obtained at 55 and 90 °C with the minimum processing time (5 min). Interestingly, the use of pure water as a solvent, which is characterized by a higher dielectric constant than less polar solvents and absorbs more microwave energy mainly in the low temperature range (25 °C–55 °C), resulted in a better extraction efficiency of phenolic compounds [97].

### 5.6. Sequential Extraction of High-Added-Value Compounds

With the aim of adopting a circular economy approach and reaching near-zero discharge, many studies have focused on the full valorization of tomato processing by-products through effective sequential extraction methods, consisting of the utilization of novel or conventional technologies in cascade, to recover the compounds of potential interest for application in different sectors.

Ouatmani et al. (2022), to increase the sustainability of industrial tomato processing waste valorization, considered the sequential recovery of oil from seeds and optimized antioxidant extraction by applying MAE. Soxhlet extraction was used to recover the oil from tomato seeds, resulting in a product rich in unsaturated fatty acids (79.83%). MAE was then applied to defatted tomato seeds to extract phenolic compounds. To identify the optimal operative conditions (solid/solvent ratio, microwave power, and processing time), the authors used RSM. The application of microwaves at 700 W for 70 s with a 32.41% ethanol concentration resulted in an amount of total phenolic compounds extracted of 268.47 mg GAE/100 g with an antioxidant activity of 84.27%. MAE enhanced the recovery of phenolic compounds compared to conventional methods, including maceration and stirring, which led to the extraction of 171.3 and 181.6 mg GAE/100 g, respectively [118].

In addition, novel technologies can be utilized in combination with each other with the aim of achieving the valorization of tomato processing by-products.

Grassino et al. (2020) investigated the combination of high hydrostatic pressure extraction (HHPE) and UAE to improve the recovery of pectin, polyphenols, and fatty acids from tomato peels. The authors reported that the application of HHPE and UAE coupled with a conventional extraction method (Soxhlet) provided an appropriate and effective solution for the successive extraction of pectin, polyphenols, and fatty acids from tomato processing by-products. Pectin extraction yield by HHPE in 45 min was 9.2%, while conventional extraction in 360 min was 7.7%. Results showed that tomato peels, both with pectin or depectinized, subjected to UAE had a high content of total phenolic compounds (TPC) (922.53–3643.88 mg/100 g), depending on the extraction time and solvent used. However, although TPC is significantly lower in depectinized residues, suggesting that a part of the phenols possibly migrated during HHPE and were released into pectin during nitric acid extraction, the amount remaining in the residual biomass is still appreciable. Interestingly, the residues obtained after UAE had high saturated FA content, such as lauric, palmitic, and stearic acids, indicating that UAE residues could be exploited for the recovery of these compounds. The authors concluded that HHPE and UAE allowed for a considerable decrease in the extraction time, and the implementation of these technologies in the extraction process can be considered an appropriate alternative to conventional methods [90].

Grassino et al. (2020) also studied other processing methods for the concurrent recovery of pectin, polyphenols, and fatty acids from tomato peels, utilizing conventional extraction methods. Before extracting polyphenols and fatty acids, pectin was recovered from tomato peels and tested as a potential tin corrosion inhibitor. The authors demonstrated that depectinized peels were effective bio-substrates for polyphenol extraction, possessing higher amounts of total phenols (2485.68–4064.46 mg/100 g) in comparison with the samples containing pectin. Additionally, depectinized samples contained more FAs (~45%) than those with pectin (~26%). Therefore, the authors demonstrated that depectinized tomato peels can be used to recover polyphenols and fatty acids, and that the pectin recovered is a more effective and efficient tin corrosion inhibitor than the apple pectin commercially available (+26%) [119].

## 6. Other Applications of Tomato Processing By-Products

### 6.1. Biofuel Production

Following the concept of near-zero waste, biorefinery has been considered, also by the European Commission, as an attractive way to valorize organic wastes and by-products and promote sustainable economic growth. These biomasses represent a cheap and abundant feedstock that could be used for energy production in anaerobic digestion (AD) plants to replace food crops [120]. Indeed, tomato processing by-products are often dumped or landfilled near processing sites, generating liquid and methane emissions due to uncontrolled anaerobic fermentation [121].

Although several research activities focused on the determination of the biogas potential of tomato processing by-products and some studies have been carried out to demonstrate their attractiveness from an energy valorization perspective, this topic is still extensively under-investigated and often contradictory findings have been achieved.

One of the main challenges for biogas production through AD is feedstock digestibility. Pre-treatment strategies, such as the utilization of chemical, thermal, biological, and physical processes to facilitate the digestion of biomass and increase the surface area accessible to microorganisms, need to be set up [122].

In this regard, Almeida et al. (2021) assessed the recovery of value-added compounds from rotten tomatoes, green tomatoes, and tomato branches through solid-liquid extraction with ethanol as a pre-treatment stage followed by biomethane production from the exhausted biomass through AD. The methane production from the untreated and the exhaust biomass was considered statistically similar (95% confidence level), with a lower value for tomato branches (141 mL CH_4_/g volatile solids) as compared to rotten and green tomatoes (232–285 mL CH_4_/g volatile solids) [123]. Allison and Simmons (2017) investigated the valorization of tomato pomace by lycopene extraction with ionic liquids and the bioconversion of the biomass to methane by AD. The authors concluded that the use of ionic liquids for the extraction of lycopene from tomato pomace was unsuitable from an AD perspective since, notwithstanding the increased digestibility during enzymatic digestion, it was hindering methane production. However, subsequent AD of the exhausted pomace indicated a slight compromise between the recovery of high-value lycopene and lower-value biogas [124].

Additionally, tomato pomace can also be successfully subjected to combined thermal and biological pre-treatments. Although hydrothermal processing and enzymatic hydrolysis led to the highest concentration of bioethanol (~20 g/L) from tomato pomace fermentation, the concentration obtained is still insufficient for forecasting industrial exploitation. In contrast, tomato pomace was a suitable biomass for acetone-butanol-ethanol-isopropanol (ABEI) fermentation, the obtained concentrations of butanol and isopropanol being only slightly lower than those of conventional industrial ABEI processes [125].

Calabrò et al. (2015) focused on the possibility of enhancing biogas production from tomato processing wastes through an alkaline pre-treatment. An average of 320 NmL/g volatile solids of methane was achieved, and no statistical differences between methane production from untreated and treated samples were detected. Therefore, tomato pomace is a suitable substrate for AD, and an alkaline pre-treatment should be considered only in those cases where the buffering capacity is insufficient for preventing the acidification of the anaerobic sludge derived from the use of tomato pomace [121].

Girotto et al. (2021) applied US to tomato pomace to achieve disintegration of the sludge undergoing AD and evaluated the effect of this pre-treatment on methane production [126]. The authors demonstrated that the highest methane production rate acquired by applying US (15 min at 152 µm amplitude) was about 90% higher than that obtained with the untreated biomass. However, an energy-oriented approach revealed that US required more energy than that generated by the increased methane yield. The process might be advantageous, also in terms of operational costs, by utilizing a pilot-scale US unit, a higher quantity of biomass, and recovering higher methane flow rates [127].

Scaglia et al. (2020) demonstrated that fiber’s biodegradability (+64%) can be increased with SFE-CO_2_, which allowed improving the characteristics of the exhausted tomato pomace, which could be considered as an effective alternative to the maize currently in use. Interestingly, for an Italian tomato cannery, a biorefinery consisting of SFE-CO_2_ + AD could lead to an additional gain of +787.9 €/ton of exhausted tomato pomace as compared to systems currently operated that do not involve the use of SFE-CO_2_ [23].

Lenucci et al. (2013) investigated the possibility of conversion of the carbohydrates present in tomato pomace and in the exhausted biomass from SFE-CO_2_ extraction of lycopene into bioethanol. The authors demonstrated, based on the results of glycosyl linkage analysis, that this technology does not affect the structure of the cell wall of polysaccharides [128].

Another factor playing a crucial role in efficient biogas production is the selection of the feedstock. Several researchers demonstrated that balancing the nutrients in co-digestion could improve the performance of the AD process. According to Szilágyi et al. (2021), tomato processing by-products could be good co-substrates with corn stover in continuous anaerobic fermentations for biogas production [129]. Likewise, Li et al. (2018) showed that the addition of 40% tomato residue to 24% corn stover and 36% dairy manure increased the methane production yield [12].

Moreover, Mahmoodi-Eshkaftaki and Ghani (2022) combined US with the co-digestion of tomato waste and cow manure to maximize bio-H_2_ and bio-CH_4_ production. Optimizing US processing conditions (197.21 W for 21.47 min) and substrate composition (96.93% tomato waste and 3.07% cow manure) improved bio-H_2_ and bio-CH_4_ production by 18.85% and 2.02%, respectively, although to avoid or decrease the formation of inhibiting compounds (maximum allowable ranges of TPC and Tannin Content are 2.1–12.5 mg/g and 5.6–25.5 mg/g, respectively), US power and sonication time must be optimized [126].

Overall, a biorefinery approach to producing a gamut of products, namely bioactive compounds and biogas, via extraction assisted process followed by AD represents a valid and effective solution enabling to maximize the yield of high-added-value compounds, and energetic content, and reduce the environmental footprint of tomato by-products [130], making such a biomass even more valuable and cost-effective than other dedicated crops [128].

### 6.2. Low-Cost Biosorbent Production

Besides the production of biogas, tomato processing residues can also be used in the production of biosorbents to reduce or eliminate contaminants, such as heavy metals, from industrial wastewater. In this regard, Azabou et al. (2020) studied the full valorization of tomato pomace by recovering natural antioxidants and edible oil from peels and seeds, using pure ethanol and hexane, respectively, and by employing the exhaust biomass to produce low-cost biosorbents. The latter, obtained from the carbonized biomass, were successfully applied for the removal of acidic dyes [131]. Likewise, Yargic et al. (2015) [132] and Heraldy et al. (2018) [133] reported that tomato wastes could be used as effective and low-cost biosorbents for the removal of copper (II) ions (92.08% removal efficiency) and Pb (II) (adsorption capacity of 152 mg/g) from aqueous solutions. Interestingly, a new composite adsorbent, applicable in the adsorption of Co (II) from water, was prepared by a chemical activation method from tomato and carrot wastes and combined with PET bottle leftovers to increase their adsorption capacity (312.50 mg/g) [134].

However, further comprehensive studies focusing on the optimization of all the biosorption parameters involved in the biosorbent production process from tomato processing residues are needed in order to eventually assess the feasibility and scalability of the process at the pilot and industrial scales.

## 7. Conclusions, Future Directions, Challenges, and Opportunities

Many researchers have pointed out the valorization of tomato processing by-products as a solution to improve the environmental sustainability and the economic performance of canning companies. Numerous valorization schemes have been proposed to explore tomato processing by-products as a suitable biomass to obtain valuable bio-based products, with the focus on recognizing the most promising high-added-value compounds to be recovered and implementing new technologies that can be complemented to the most traditional methods, solid-liquid extraction among others, to improve mass transfer and extraction yield, consume lower quantities of solvents, and reduce energy consumption. However, in the suggested approaches, some challenges can still be envisaged, and further research efforts should be made, as discussed in the following.

### 7.1. Co-Extraction of the Target Compounds in a Cascade Approach

The cascade extraction method has been identified as one of the most promising pathways to achieve the near-zero waste concept and accelerate the transition of the tomato processing industry to a circular bioeconomy. However, due to the simultaneous extraction of high-added-value compounds, the cascade extraction approach reported and discussed in the literature is under question. The main challenge in this regard could be the selection of appropriate new technologies and extraction procedures that assure the minimum interactions between the different steps and enable maximizing the extraction yield and purity of the extracts. However, although introducing cutting-edge technologies, such as HPH, PEF, SFE-CO_2_, UAE, and MAE, may have a beneficial effect on food waste valorization, their inherent limitations could represent significant barriers to their use in actual processing lines. More work is needed to exploit the use of these technologies on an industrial scale, including their integration in cascade processes. The use of innovative combined technologies for agri-food waste valorization could improve their respective strengths, including lower operative costs compared to conventional processes, and offset their disadvantages, mainly the high investment costs, through the total reuse of a cheap source to recover valuable compounds of natural origin.

### 7.2. Upscaling of Tomato Waste Valorization

Most of the studies reported in the literature discuss the results of experiments carried out at the lab-scale and pilot-scale. Very rarely is information on experiments at the industrial scale provided, including the technical and economic feasibility of full-scale operations. To overcome these problems, it would be preferable to generalize and simplify the scheme of the valorization process. Therefore, further research efforts are needed for the optimal selection of the alternative technologies to be potentially used for waste valorization and to set up design criteria for the industrial up-scaling based on the results of laboratory-scale experiences. The transition from waste to wealth implies high costs for research and development. Hence, to justify the investments, it is essential to define wider research plans and carry out the related activities aiming to individuate all the intermediate products of the different recovery steps and determine the potential for the results exploitation. Furthermore, efforts are also needed to localize the producers, explore the possibility of creating new knowledge-based industrial initiatives for agri-food waste valorization, and better identify the potential end users.

### 7.3. Environmental Evaluation of Processing Methods for Agri-Food Waste Valorization

To prevent worsening the problems of agro-industrial waste disposal and the increase of their environmental footprint, an in-depth assessment of the sustainability of the extraction procedures of the compounds of interest should be performed. In fact, the extraction of high-added-value compounds from wastes involves the use of potentially polluting solvents that, in a rather small amount or in traces, remain trapped in the exhausted biomass. This causes an additional environmental impact when biomass is reused or landfilled. Hence, the advantages of utilizing tomato waste as secondary feedstock to recover target compounds should not be hindered by the environmental burdens of the valorization processes envisaged. Therefore, it is necessary to deeply examine the problem and properly design the bio-refinery procedures using environmental evaluation techniques that consider the comprehensive by-product life cycle, identify the most appropriate novel technologies for the pre-treatment of the biomass, and individuate the proper solvents for solid-liquid extraction if their use is unavoidable and that cause very limited detrimental effects to the environment. The selection of greener extraction strategies represents a suitable and valuable approach to turning low-value wastes into high-value ingredients.

### 7.4. Adaptation of Appropriate Pre-Treatment Stages in Combination with Green Solvents for Agri-Food Waste Valorization

The identification of suitable alternative methods that are safe, solvent-free, and environmentally friendly to recover target compounds with high extraction yields can be considered a frontier for promoting and spreading the valorization of agri-food biomass and achieving the goal of zero waste production. Novel technologies for biomass pre-treatments have been recently proposed to complement traditional solid-liquid extraction and overcome the above-mentioned drawbacks. PEF assisted extraction, UAE, and MAE of target compounds from agri-food wastes allowed reducing the extraction times and increasing the extraction yields, but the downstream processing of the extracts still involve the use of solvents. Therefore, the selection of the most suitable novel technologies should consider the advantages and limitations associated with their implementation in the process scheme, making them appropriate for obtaining specific effects on the biomass and recovering target compounds. In particular, the utilization of non-thermal, non-destructive, and selective technologies, such as PEF and SFE-CO_2_, that cause only limited isomerization and degradation of bioactive compounds, can be suggested to recover thermosensitive compounds, such as lycopene, resulting in a higher quality and purity of the lycopene compared to that recovered by applying other technologies. HPH, UAE, and MAE are suitable to unlock also complex and high molecular weight polysaccharides, namely pectin and proteins from tomato pomace and peels and facilitate the extraction of oil from tomato seeds. Nonetheless, HPH is less suitable for processing lignocellulosic materials such as tomato seeds, which could easily cause obstruction or blockage of the micrometric valve. Given the current knowledge on innovative extraction procedures for agri-food waste valorization, the technologies and schemes proposed can be exploited only as soon as the data available is confirmed at a larger scale, and selective and low-cost green solvents will be proven effective when coupled with the physical methods proposed to increase the effectiveness of the recovery process. It should also be demonstrated more clearly, by providing a more meaningful number of consistent and robust data, that the use of these complex extraction procedures will enable the use of simplified, less severe, and less expensive downstream processes, and that the environmental and economic sustainability of waste valorization will be ensured.

### 7.5. Upstream Process

To promote the exploitation of agri-food biomass on an industrial scale, further information is still necessary, and a multidisciplinary approach must be identified to obtain a range of compounds that can remain stable during the shelf life, do not adversely affect the workability of the products when used as ingredients, and retain their characteristics in the final products. Further studies are also needed on the stabilization of compounds, the design of bioactive particles with tailored characteristics (particle size, hydrophilicity, and hydrophobicity) that can be easily incorporated into the product formulation, withstand the production process without losing stability and bioavailability and, in case of incorporation in foods, withstand the passage into the gastrointestinal tract (GI) to be properly delivered into the target organ and exercise their beneficial health effects.

In conclusion, the selection of environmentally friendly and optimized extraction procedures for the recovery of high-added value compounds from agri-food wastes, even using a cascade approach, can be only possible if process upscaling criteria, the identification of appropriate pre-treatments, upstream processes, and the evaluation of environmental and economic impacts of the bio-refinement processes are provided. This represents the real challenge also for tomato wastes and by-product valorization. Moreover, further efforts should be made in individuating broader utilization of the gamut of products recovered, which would further support the implementation of novel processes in an industrial environment.

## Figures and Tables

**Figure 1 foods-12-00166-f001:**
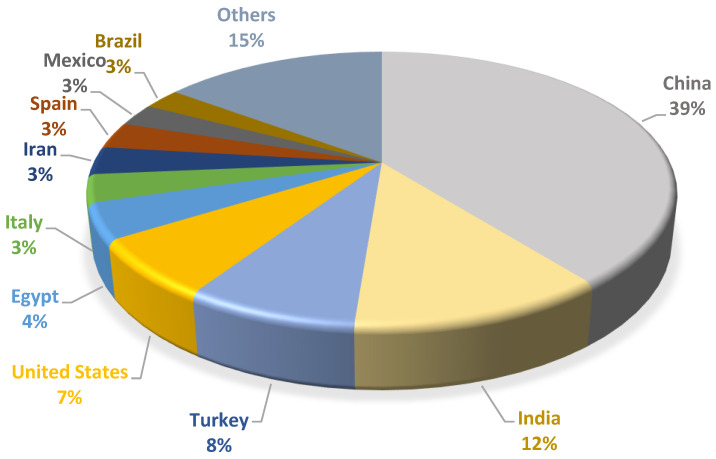
Top ten countries for tomato production based on FAOSTAT (2019) [8].

**Figure 2 foods-12-00166-f002:**
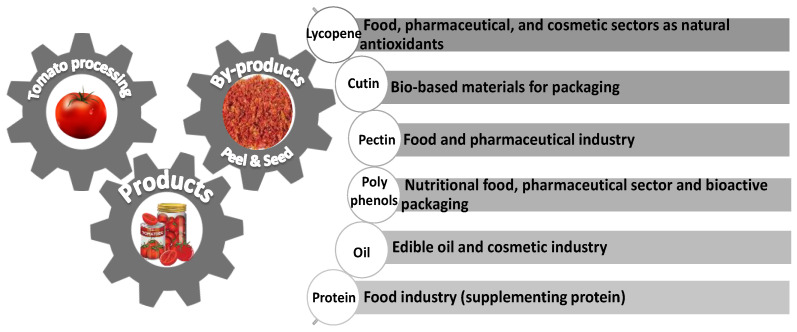
Schematic of tomato processing by-product valorization, compounds recovered, and their potential fields of application.

**Figure 3 foods-12-00166-f003:**
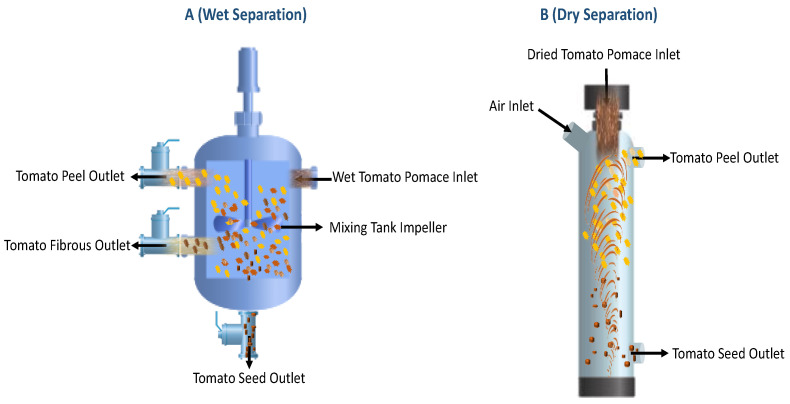
Schematization of wet (**A**) and dry (**B**) methods for tomato peel and seed separation.

**Figure 4 foods-12-00166-f004:**
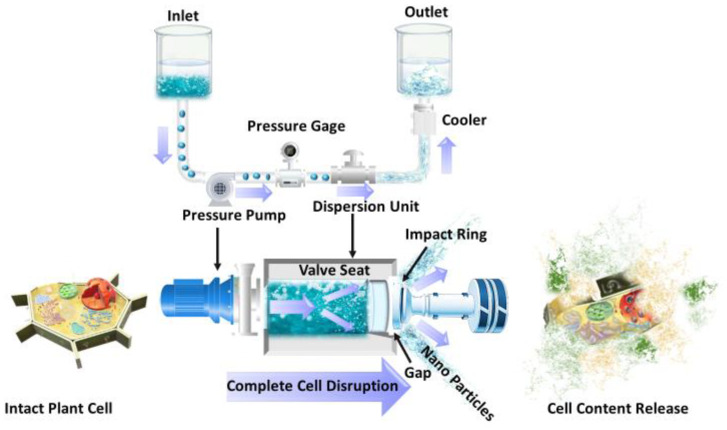
Schematic representation of HPH technology and its effect on plant cell.

**Figure 5 foods-12-00166-f005:**
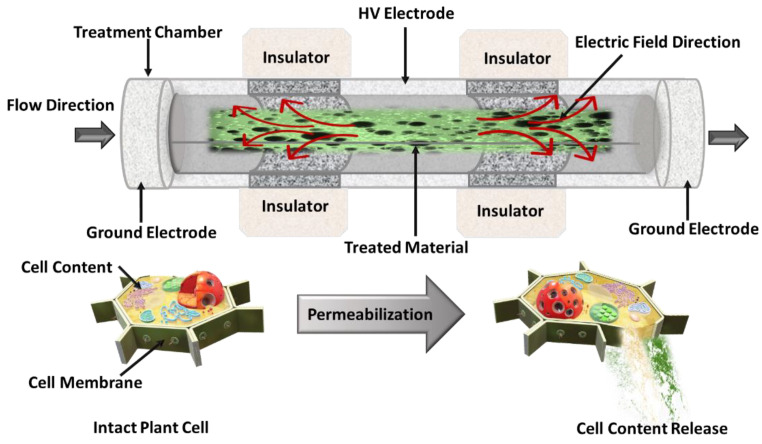
Schematic representation of a continuous co-filed PEF treatment chamber.

**Figure 6 foods-12-00166-f006:**
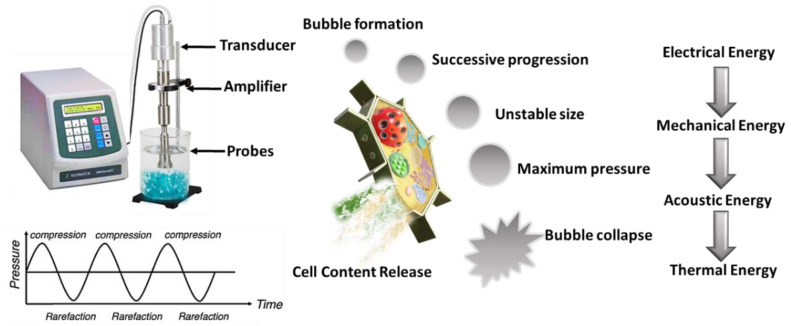
Schematic representation of ultrasound technology and its effect on plant cells.

**Figure 7 foods-12-00166-f007:**
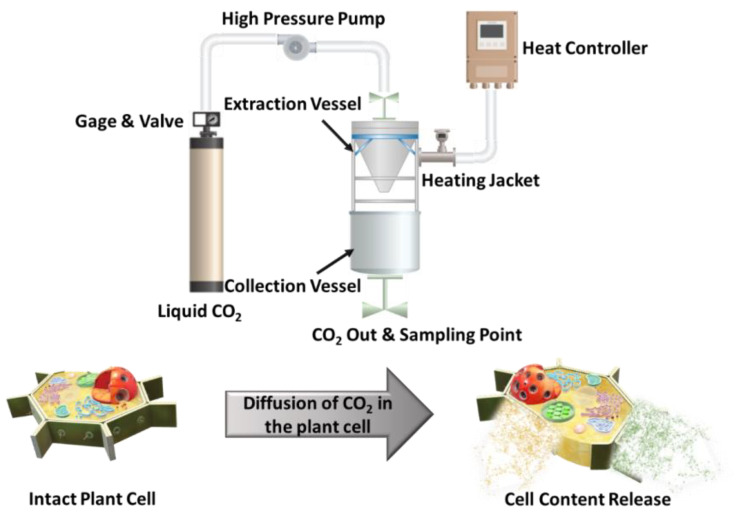
Schematic representation of a supercritical carbon dioxide extraction system.

**Figure 8 foods-12-00166-f008:**
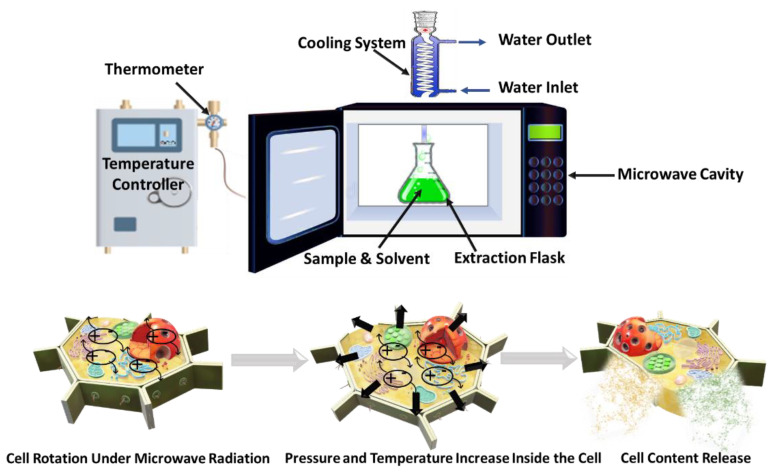
Schematic of microwave technology and its mechanism.

**Table 1 foods-12-00166-t001:** Main compounds in tomato pomace, peels, and seeds.

Material	Fibres [g/100 g dw]	Proteins [g/100 g dw]	Oil [g/100 g dw]	Lycopene × 10^3^ [g/100 g dw]	References
Pomace	39.11–59.03	15.08–24.67	2.00–16.24	9.82–611.105	[27,28,29,30]
Peels	62.79–78.56	1.85–11.13	1.63–5.50	50–1930	[15,31,32,33,34]
Seeds	16.00	20.2–40.94	17.80–24.50	22.01–37.43	[19,29,35,36,37]

**Table 2 foods-12-00166-t002:** Carotenoids composition in tomato peels [29].

Lycopene [g/100 g of TC]	Phytoene [g/100 g of TC]	Phytofluene [g/100 g of TC]	β-Carotene [g/100 g of TC]	Cis-Lycopene [g/100 g of TC]	Lutein [g/100 g of TC]
86.12	3.15	2.31	2.11	1.71	1.51

TC: total carotenoids.

**Table 3 foods-12-00166-t003:** Emerging technologies used for bioactive compound extraction from tomato processing by-products, type of biomass, target functional compounds recovered, experimental conditions, and main research findings.

Technology	Material	Target	Experimental Condition	Optimal Condition	Main Research Findings	References
HPH ^1^	Tomato peels	•Lycopene •Polyphenols •Proteins	•Number of passes: 1–10 •P: 100 MPa •Solvent: Water	•P: 100 MPa •Number of passes: 10	•Proteins (+70.5%) •Polyphenols (+32.2%) •Extracts antioxidant activity (+23.3%) •Lycopene recovered (up to 56.1% of that initially present in the peels)	[15]
PEF ^2^	Industrial tomato waste	•Lycopene	•E: 1–5 kV/cm •W_T_: 5–10 kJ/kg •Solvent: Acetone, Ethyl lactate	•E: 5 kV/cm •W_T_: 5 kJ/kg • Acetone	•Extraction rate (+27–37%) •Lycopene recovery yield (+12–18%) •Antioxidant power (+18.0–18.2%)	[11]
	Industrial tomato peels	•Lycopene •Proteins •Phenolic compounds	•E: 1.0–5.0 kV/cm •W_T_: 5.7–22.8 kJ/kg •n: 0–500 pulses	•PEF: 2 kV/cm for 700 pulses •PEF: 5 kV/cm for 1.5 µs •PEF: 1.0 kV/cm for 7.5 µs	•Carotenoid extraction yield (+56.4%) •Concentration of total phenolic compounds (+56.16 mg gallic acid/kg) •Protein recovery (1.45 mg/g tomato waste) •Lycopene recovery (0.143 mg/g tomato waste)	[83]
	Tomato peels	•Lycopene	•E: 0.5–5 kV/cm •W_T_: 0.5–20 kJ/kg •Solvent: Acetone •T: 20–50 °C	•E: 5 kV/cm •W_T_: 5 kJ/kg •Solvent: Acetone •T: 50 °C	•Total carotenoids (+47%) • Antioxidant power (+68%)	[84]
	Tomato pulp and peels	•Lycopene	•E: 3–7 kV/cm •W_T_: 0.54 to 13.50 kJ/kg •n: 5 to 100 pulses of 3 µs •Solvent: Hexane: Ethanol: Acetone	•E: 5 kV/cm •30 pulses of 3 µs •Solvent: Hexane: Ethanol: Acetone (50:25:25)	•Total carotenoids (+39%) •Reduced hexane consumption (from 45 to 30%) without changing the carotenoid extraction yield	[85]
SB ^3^—PEF	Tomato peels	•Lycopene	•E: 0.25–0.75 kV/cm •W_T_: 1 kJ/kg •Time: 1 min •T: 50–70 °C •Solvent: Acetone	•E: 0.5 kV/cm •W_T_: 1 kJ/kg •T: SB at 60 °C	•Carotenoid content 0.379 mg/g fw	[17]
UAE ^4^	Tomato pomace	•Pectin	•Frequency: 37 kHz •Time: 15, 30, 45, 60, and 90 min •T: 60 °C and 80 °C •Solvent: Ammonium oxalate/oxalic acid	•Frequency: 37 kHz •Sonication time: 90 min, •T: 80 °C	•Pectin yield 31.2% by CE (1440 min) and 36% by UAE (15 min)	[86]
	Industrial tomato waste	•Lycopene	CSE •T: 20, 40, and 60 °C •Time: 10, 20, 30 and 40 min •Solvent: Hexane: Acetone: Ethanol (2:1:1) UAE • Power: 50, 65 and 90 W •Time: 1, 2, 5, 10, 15, 20 and 30 min	•Solvent/solid ratio: 50:1 (*v*/*w*) •T: 60 °C •Time: 40 min •Solvent/solid ratio: 35:1 (*v*/*w*) •Power: 90 W •Time: 30 min	•To recover about 80% of lycopene UAE extraction time was 10 min, CSE extraction time at least 20 min •UAE of lycopene requires lower time, temperature, and amount of solvent than CSE	[87]
	Tomato Pomace	•Lycopene	•T: 40, 55, 70 °C •Solvent mixture: 30, 65, 100 *v*/*v* •Solvent/solid ratio: 50, 75, 100 •Extraction time: 20, 35, 50 min •Frequency: 40 kHz •Power: 100 W	•T: 63.4 °C •Proportion of ethyl acetate in solvent mixture: 30% *v*/*v* •Solvent/solid ratio: 100 mL/20 g •Time: 20 min	•UAE lycopene recovered 1.33 mg/g dw (+9.4%) higher than that of CSE (1.209 mg/g dw)	[88]
	Tomato Pomace	•Lycopene	•T: 40, 55, 70 °C •Time: 20, 35, 50 min • Amplitude: 20,30,48, 65% • Time: 30, 50, 70 s •V: 32, 45, 60, 90 mL	•T: 65 °C •Time: 20 min • Solvent/solid ratio: 72 mL/g •Amplitude: 65% •Time: 33 s •V: 90 mL	•UAE lycopene recovered 1.53 mg/g	[89]
HPPE ^5^ + UAE + CSE ^6^ (Soxhlet)	Tomato pomace	•Pectin •Polyphenols •Fatty acid	HHPE •T: 80 °C •P: 300 MPa •Time: 10, 20, 30, and 45 min •Solvent: Nitric acid CSE • Solvent: Mixture chloroform and methanol (50:50, *v*/*v*) at solvent boiling temperature		•Simultaneous extraction of different compounds, (decreased yield of individual compounds) •HHPE, UAE, and CSE used in combination allow decreasing the extraction time	[90]
UAE MAE ^7^ OHAE ^8^ UAME ^9^ UAOHE ^10^	Tomato peels	•Pectin	•Sonication power: 450, 600 and 750 W •Time: 2, 4, 6, 8, 10, 12, 14, and 16 min •Microwave power: 540 W applied for 4 min	•US (450 W, 8 min) •MW (540 W)	•OHAE pectin extraction yield increased 9.30% •MAE pectin extraction yield increase of 25.42% •UAME can be used as an efficient pectin extraction technique from tomato by-products	[91]
UAE	Tomato seeds	•Oil	•Power: 550 W, 37 kHz •Time: 30, 60, and 90 min • T: 70 °C •Solvent: Hexane •Water immersion: 25–40 °C	•Power: 550 W, 37 kHz •Time: 90 min •T: 40 °C	•UAE oil extraction yield increases up to 28.11% (15.91% more than that of untreated samples)	[92]
SFE ^11^ CO_2_	Industrial tomato peels by-product	•β-carotene •Lycopene	•T: 50–80 °C •P: 30–50 MPa •Flow rates: 3–6 g CO_2_/min Time: 105 min	•P: 40 MPa •T: 80 °C •Flow rates: 4 g CO_2_/min •Time: 105 min	•Extraction yield of 28.38–58.8% for β-carotene, and 32.02–60.85% for lycopene •Lycopene recovered 0.728 mg/g dw	[93]
	Tomato peels	•Lycopene	•T: 70, 74, 80 °C • P: 20, 40 MPa •Time: 155 min	•Time: 155 min •P: 40 MPa •T: 74 °C	•Lycopene recovered 5.28 mg/g dw	[94]
	Tomato Pomace	•Oil	•T: 40, 50, 60, 80 °C •Tim: 2–8 h •P: 210–280 bar	•P: 280 bar •T: 40 °C •Tim: 2.5 h	•Yield of tomato seed oil was 0.25 g/g (solubility 14 mg/dm^3^)	[95]
	Tomato pomace	•Lycopene	•P: 30–50 MPa •T: 40–80 °C •Peel/seed ratio: 30/70 and 70/30	For peel/seed ratio: 70/30 •P: 50 MPa •T: 80 °C For peel/seed ratio: 30/70 • P: 50 MPa •T: 60 °C	•Maximum lycopene recovered: 0.358 mg of extract/kg of raw material (measured) and 0.320 mg of extract/kg of raw material (predicted) •Lycopene recovery is affected by peel/seed proportion, pressure, and temperature.	[96]
MAE	Tomato peels Tomato peels	•Phenolic compounds •Lycopene •Beta-carotene	•T: 25, 55, and 90 °C •Time: 5 and 10 min • Solvent: Methanol and HCl •Time: 30, 60, and 90 s •Power: 180, 300, and 450 W	•1% HCl to 50 or 70% methanol for phenolic acids T: 90 °C •MAE condition is 300 W for 60 s	•The average total phenolic content 0.053 g/mg •Extraction time does not affect TF, TP, and phenolic compound recovery, and temperature and solvent have a significant effect on polyphenols yield •Lycopene recovered 0.0574 mg/g dw •Beta-carotene recovered 0.0483 mg/g dw	[97]
UAE MAE	Tomato seeds	•Oil	•Pre-processing with hot water 40 °C for 24 h and UAE 550 W, 37 kHz •T: 25–40 °C •Time: 30 and 60 min •Extraction for MAE: powers of 250 and 600 W Time: 90 min Solvent: Hexane	•Ultrasound: 60 min •Microwave: 600 Watts •T: 40 °C •Pre-processing with hot water 40 °C for 24 h	•Extraction efficiency up to +23.03% •Extraction time decrease of 1.5 min with MAE and 30 min with UAE	[98]

^1^. HPH, high-pressure homogenization ^2^. PEF, pulsed electric field ^3^. SB, steam blanching ^4^. UAE, ultrasound-assisted extraction ^5^. HPPE, high pressure processing extraction ^6^. CSE, conventional solvent extraction ^7^. MAE, microwave-assisted extraction ^8^. OHAE, ohmic heating-assisted extraction ^9^. UAE-MAE, ultrasound-assisted extraction-microwave-assisted extraction ^10^. UAOHE, ultrasound-assisted ohmic heating extraction ^11^. SFE, supercritical fluid extraction.

**Table 4 foods-12-00166-t004:** Main features, advantages, and limitations of emerging extraction technologies.

Emerging Technologies	Main Features	Advantages	Limitations	References
HPH	Using high pressure intensifiers to expose biomass to high-levels of mechanical stress and shear results in complete deformation and disruption of the plant cell structure and improves the release of intracellular bioactive compounds from agri-food by-products	Short extraction time No solvent or a small amount of solvent is required Environmentally friendly method Improved extraction yield	Non-selective method High costs and capital investments Operators training is required	[15,99,100]
PEF	Exposing plant matrices to a moderate electric field and relatively low energy input induces the electropermeabilization of cell membranes by pore formation	Selective extraction of compounds Energy efficient and low-cost operation Short processing time Non-thermal, and non-destructive technology Continuous operability Easy scalability at industrial level	High costs and capital investments Operator training is required Reduced uniformity of PEF treatment due to the presence of air bubbles Uneven distribution of the electric field in the treatment chamber that can be corrected by geometry, insulator design, or inserting metal mesh Arching phenomenon and undesirable electrochemical reactions due to high electric field intensity	[101,102,103]
US	Acoustic cavitation followed by the release of a huge amount of energy creating shear stresses, allowing greater penetration of the solvent into the plant tissue	Low energy requirement Short extraction time Less solvent requirement Improved extraction efficiency	Non-selective method Damages to heat labile compounds Decreased intensity of equipment due to aging, lessening the reproducibility	[22,91,92]
SFE-CO_2_	Supercritical fluids allow for increased solvating power of gases beyond their critical point to extract compounds from the biomass	Low temperature operation Recovery of thermosensitive compounds Selectivity increases with changing pressure and temperature Recovery of extracted compounds with little or no solvent residues by depressurization Easy scalability of the process at the pilot and industrial level Reuse supercritical carbon dioxide Use of environmentally friendly solvents	High capital investments Complexity of the system Operators training is required Poor selectivity for polar compounds due to the low polarity of supercritical carbon dioxide	[93,94,104]
MW	Microwave heating causes physical and biological modifications of the biomass, improving the penetration of the extracting solvent into the vegetable tissue	Short extraction time High extraction yield Energy efficient process Low capital investments Low environmental pollution	Non-selective method Not uniform heating, reducing extraction efficiency Thermal degradation of phenolic compounds due to overheating of biomass Limited penetration of microwaves for scaling up Changes induced on the chemical structure of the target compounds, hindering their bioactivity and reducing their potential applications Limitation for the recovery of nonpolar compounds	[82,98,105,106]

## Data Availability

No new data were created or analyzed in this study. Data sharing is not applicable to this article.

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
