# Peer review of "A Comprehensive Overview of Tomato Processing By-Product Valorization by Conventional Methods versus Emerging Technologies"

_foods, 2022, doi:10.3390/foods12010166_

Round 1

Reviewer 1 Report

Just minor revisions:

- Please revise the FAO, OCDE, and WEF data, you need update data about Mexico, Brasil and other in Fig 1.

- You need an update of references about harnessing of tomato byproducts.

- Please revise the methodology, this need be so clear, such that anyone can repeat your described procedures.

- Revise the discusion and conclusion, these need be so clear.

Author Response

Reviewer 1

Comments to the Authors

- Please revise the FAO, OCDE, and WEF data, you need update data about Mexico, Brasil and other in Fig 1.

We thank the reviewer for the comment. The updated data for tomato production have been reported in the manuscript (Figure 1) based on the most updated data from FAO, including those of Mexico and Brazil, among others. Nevertheless, the annual tomato production in Italy has been updated to the year 2022.

The new references have been added in the reference list accordingly.

- You need an update of references about harnessing of tomato byproducts.

We thank the Reviewer 1 for his comment. The updated information about tomato processing by-products have been added to the review (Lines 71 – 72). Moreover, recent references focusing on the valorisation of tomato processing by-products and on the characteristics and applications of the recovered bioactive compounds, have been added throughout the manuscript (References: 46, 47, 54, 58, 71, 113).

- Please revise the methodology, this need be so clear, such that anyone can repeat your described procedures.

The authors would like to thank the Reviewer 1 for his comment. However, this manuscript is not a research paper, where the experimental protocols necessary to obtain the data need to be clearly described. This paper discusses the most recent advancements in tomato processing by-products valorisation for the recovery of different valuable compounds. Thus, the procedures and methodologies reported are those described in previous studies available in the literature and as such briefly summarized and discussed in our review paper. Nevertheless, some clarification on the procedures reported in the manuscript have been reported accordingly.

- Revise the discussion and conclusion, these need be so clear.

We thank the reviewer for the comment. The discussion and conclusions have been revised, some concepts have been rephrased and explained clearly throughout the manuscript.

Reviewer 2 Report

This manuscript is a review about tomato byproducts and new technologies to valorize them. I just have one request: explore more the use of green solvents, such as bio-based solvents, ionic liquids and eutectic solvents in the recovery of tomato metabolites

Author Response

Reviewer 2

Comments to the Authors

This manuscript is a review about tomato byproducts and new technologies to valorize them. I just have one request: explore more the use of green solvents, such as bio-based solvents, ionic liquids and eutectic solvents in the recovery of tomato metabolites.

The authors would like to thank the Reviewer 2 for his constructive comment. More studies on the use of green solvents in the recovery of tomato metabolites have been added and discussed (lines 293-303, and lines 693-704). Moreover, the new references (54, 113) have been added to the reference list accordingly.

Reviewer 3 Report

This paper aimed to propose a valorisation of tomato processing by-products with a specific focus on the use of “green technologies”, including High-Pressure Homogenization (HPH); Pulsed Electric Fields (PEF); Supercritical Fluid (SFE-CO2); Ultrasounds (UAE); and Microwaves (MAE); suitable to maximize the extraction yields of the bioactive compounds from tomato residues, reducing the solvents consumption and the extraction time, to boost the recovery of bioactives from tomato processing by-products.

Abstract and title

Abstract is concise and clearly written, with a good command of English, and clear representation of the aim of the review paper. Furthermore, it is adequately structured: background of the proposed research, methods used, and main conclusions were mentioned.

The title of the paper adequately reflects the subject under investigation in the proposed study.

2. Chemical composition and characteristics of tomato pomace, peels, and seeds

The authors clearly represented the importance of the issue described followed by schematic figure of tomato processing by-products valorisation which helps that the end of the section is clear and concise.

3. Separation of peels and seeds from tomato pomace

In this section is given comprehensive overview regarding wet and dry separation methods of peels and seeds from tomato pomace with all  advantages and disadvantages.

Suggestion: as in previous chapter should be followed by schematic figure for wet and dry separation

4. Recovery of high-added-value compounds from tomato processing by-products by conventional methods

In this section is very concise and extensive descibed lycopene, cutin, pectin, proteins characteristics, applications and extraction and sections have been shown jointly.

5. Application of green technologies for tomato processing by-products valorisation

In this section have been shown jointly high pressure homogenization (HPH) technology, pulsed electric fields (PEF) technology, ultrasound technology, supercritical fluid extraction microwave technology, with their advantages compered to conventional methods, with theoretically explanantion of every individal tretman followed by schematic representation

This is the comprehensive overview with an environmentally friendly method and perspective energy consumption summary, with excellent global production prediction compounds from tomato processing by-products, the safety of the extracted compounds is highlighted, with a review on circular economy approach and following the concept of near-zero waste.

However, the paper lacks the comparison with previous studies.

Author Response

Reviewer 3

Comments to the Authors

This paper aimed to propose a valorisation of tomato processing by-products with a specific focus on the use of “green technologies”, including High-Pressure Homogenization (HPH); Pulsed Electric Fields (PEF); Supercritical Fluid (SFE-CO2); Ultrasounds (UAE); and Microwaves (MAE); suitable to maximize the extraction yields of the bioactive compounds from tomato residues, reducing the solvents consumption and the extraction time, to boost the recovery of bioactives from tomato processing by-products.

Abstract and title

Abstract is concise and clearly written, with a good command of English, and clear representation of the aim of the review paper. Furthermore, it is adequately structured: background of the proposed research, methods used, and main conclusions were mentioned.

The title of the paper adequately reflects the subject under investigation in the proposed study.

  1. Chemical composition and characteristics of tomato pomace, peels, and seeds

The authors clearly represented the importance of the issue described followed by schematic figure of tomato processing by-products valorisation which helps that the end of the section is clear and concise.

  1. Separation of peels and seeds from tomato pomace

In this section is given comprehensive overview regarding wet and dry separation methods of peels and seeds from tomato pomace with all  advantages and disadvantages.

Suggestion: as in previous chapter should be followed by schematic figure for wet and dry separation.

The authors would like to thank the Reviewer 3 for this suggestion. A schematization of the wet and dry separation methods for tomato peels and seeds has been presented in Figure 3.

  1. Recovery of high-added-value compounds from tomato processing by-products by conventional methods

In this section is very concise and extensive descibed lycopene, cutin, pectin, proteins characteristics, applications and extraction and sections have been shown jointly.

  1. Application of green technologies for tomato processing by-products valorisation

In this section have been shown jointly high pressure homogenization (HPH) technology, pulsed electric fields (PEF) technology, ultrasound technology, supercritical fluid extraction microwave technology, with their advantages compered to conventional methods, with theoretically explanantion of every individal treatment followed by schematic representation

This is the comprehensive overview with an environmentally friendly method and perspective energy consumption summary, with excellent global production prediction compounds from tomato processing by-products, the safety of the extracted compounds is highlighted, with a review on circular economy approach and following the concept of near-zero waste.

However, the paper lacks the comparison with previous studies.

The authors thank the Reviewer 3 for his comment.

However, a table (Table 3) that summarized the main research findings regarding the application of emerging technologies for bioactive compounds extraction from tomato processing by-products, type of biomass, target functional compounds recovered, experimental conditions, is reported and discussed in the manuscript. Nevertheless, new updated references have been added in the paper according to the reviewer suggestions (lines 293-303, and lines 693-704).